# Structural basis for Scc3-dependent cohesin recruitment to chromatin

Yan Li[1], Kyle W Muir[1†]*, Matthew W Bowler[1], Jutta Metz[2,3], Christian H Haering[2,3], Daniel Panne[4]*

[1]European Molecular Biology Laboratory, Grenoble, France; [2]Cell Biology and Biophysics Unit, European Molecular Biology Laboratory, Heidelberg, Germany; [3]Structural and Computational Biology Unit, European Molecular Biology Laboratory, Heidelberg, Germany; [4]Leicester Institute of Structural and Chemical Biology, Department of Molecular and Cell Biology, University of Leicester, Leicester, United Kingdom

**Abstract** The cohesin ring complex is required for numerous chromosomal transactions including sister chromatid cohesion, DNA damage repair and transcriptional regulation. How cohesin engages its chromatin substrate has remained an unresolved question. We show here, by determining a crystal structure of the budding yeast cohesin HEAT-repeat subunit Scc3 bound to a fragment of the Scc1 kleisin subunit and DNA, that Scc3 and Scc1 form a composite DNA interaction module. The Scc3-Scc1 subcomplex engages double-stranded DNA through a conserved, positively charged surface. We demonstrate that this conserved domain is required for DNA binding by Scc3-Scc1 in vitro, as well as for the enrichment of cohesin on chromosomes and for cell viability. These findings suggest that the Scc3-Scc1 DNA-binding interface plays a central role in the recruitment of cohesin complexes to chromosomes and therefore for cohesin to faithfully execute its functions during cell division.
DOI: https://doi.org/10.7554/eLife.38356.001

*For correspondence:
kmuir@mrc-lmb.cam.ac.uk (KWM);
panne@embl.fr (DP)

Present address: †MRC Laboratory of Molecular Biology, Cambridge, United Kingdom

Competing interests: The authors declare that no competing interests exist.

## Introduction

To ensure that each daughter cell receives an equal complement of genetic information, cognate chromatids are paired through replication-coupled sister chromatid cohesion. Cohesion is then actively maintained and eventually enables attachment of kinetochores to the mitotic spindle micro-tubules emanating from opposite poles to ensure chromosome bi-orientation, prior to subsequent segregation of sister chromatids into daughter cells (*Nasmyth and Haering, 2009*; *Peters and Nishiyama, 2012*).

Cohesion is facilitated by cohesin, a member of the Structural Maintenance of Chromosomes (SMC) family of protein complexes, which is responsible for genome organisation across all domains of life (*Palecek and Gruber, 2015*; *Wells et al., 2017*). Cohesin complexes form tripartite rings, comprising Smc1-Smc3 and the kleisin subunit Scc1, that are proposed to topologically entrap sister DNA molecules (*Gligoris et al., 2014*; *Nasmyth and Haering, 2009*).

The chromosomal addresses of cohesin loading are determined by the Scc2-Scc4 complex, which is enriched at centromeres via direct contacts with kinetochore proteins and promotes DNA-stimulated ATP hydrolysis by the Smc1-Smc3 ATPase heads to drive chromatin entrapment (*Ciosk et al., 2000*; *Hinshaw et al., 2017*; *Murayama and Uhlmann, 2014*). Conversely, dynamic release of DNA from the ring is achieved either by the proteolytic cleavage of the Scc1 kleisin subunit by separase protease (*Uhlmann et al., 2000*), or by the opening of an 'exit gate' formed at the Scc1 and Smc3 interface. Release of the latter is inhibited by Smc3 acetylation and is controlled by the accessory

factors Scc3, Wapl and Pds5 (*Beckouët et al., 2016*; *Rolef Ben-Shahar et al., 2008*; *Rowland et al., 2009*; *Unal et al., 2008*).

Cohesin can associate dynamically with chromatin through alternating cycles of DNA entrapment and release until ring opening is inhibited by acetylation of the Smc3 ATPase head. At this point, DNA is thought to remain topologically entrapped within the SMC-kleisin ring (*Gligoris et al., 2014*; *Beckouët et al., 2016*). A prevailing model of genome organisation posits that chromatin loops entrapped within the lumen of the SMC ring are then processively enlarged by extrusion, which presumably forms the basis for higher-order chromatin structure (*Alipour and Marko, 2012*).

Whereas the essential role of cohesin in modulating genome architecture, gene expression and DNA damage repair is increasingly well established, comparatively little information is available concerning the molecular determinants of its association with chromatin (*Busslinger et al., 2017*; *Haarhuis et al., 2017*; *Schwarzer et al., 2017*; *Wu and Yu, 2012*). Considerable advances have been made in dissecting the compositional and biochemical prerequisites for topological entrapment of DNA by the cohesin complex (*Davidson et al., 2016*; *Kanke et al., 2016*; *Stigler et al., 2016*; *Murayama and Uhlmann, 2014*; *2015*). However, how cohesin engages its DNA substrate and how such interactions might be regulated is yet to be fully elucidated. Recent work identified a direct DNA-binding site in the paralogous condensin complex, in which the HEAT repeat subunit Ycg1, in complex with the kleisin subunit Brn1, contacts the DNA double helix backbone and stabilizes its association through DNA entrapment within the Brn1 peptide loop (*Kschonsak et al., 2017*).

To investigate whether cohesin can similarly establish direct contacts with DNA, we produced a series of structurally well-defined globular cohesin subcomplexes and systematically interrogated their ability to bind to DNA. We found that the essential HEAT-repeat subunit Scc3, when in complex with Scc1, binds double-stranded DNA (dsDNA). We further determined the crystal structure of the yeast Scc3-Scc1 complex bound to DNA, which revealed that DNA binding is mediated by a composite surface comprising positively charged amino acid residues from Scc3 and Scc1. Charge-reversal mutagenesis of this interface demonstrates that DNA substrate engagement by the Scc3-Scc1 complex is essential for cohesin to bind to DNA in vitro or chromosomes in vivo and is therefore indispensable for cell viability. Thus, direct DNA substrate engagement through the newly discovered DNA-binding interface of the Scc3-Scc1 subcomplex is a key element for the cohesin cycle.

## Results

To investigate the DNA-binding properties of cohesin, we co-expressed and purified defined globular domains and subcomplexes of the cohesin complex from the budding yeast *Saccharomyces cerevisiae* (*Figure 1A*, *Figure 1—figure supplement 1*, *Figure 1—figure supplement 2*). These encompassed the Smc3 ATPase head domain bound to an N-terminal fragment of Scc1 (Smc3hd-NScc1), the Smc1 ATPase head domain bound to a C-terminal fragment of Scc1 (Smc1hd-CScc1), as well as an Scc3-Scc1 subcomplex (Scc3T-Scc1K) (*Figure 1B*). In addition, we produced an Smc1-Smc3 hinge heterodimer, Pds5 bound to a Scc1 fragment (*Muir et al., 2016*) as a full-length (Pds5fl) or truncated variant (Pds5T), as well as Wapl as full-length (Waplfl) or truncated variants (WaplC; *Figure 1—figure supplement 1*). Consistent with prior studies (*Murayama and Uhlmann, 2014*), we found that the Scc3T-Scc1K subcomplex and the Smc1-Smc3 hinge heterodimer bound DNA, as seen by the appearance of slower-migrating species in electrophoretic mobility shift assays (EMSAs), as did the Smc3hd-NScc1 module, which has previously been implicated as a DNA sensor in cohesin (*Murayama and Uhlmann, 2014*) (*Figure 1C*, *Figure 1—figure supplement 2*). As expected for non-sequence specific DNA-binding factors, longer DNA fragments (>21 base pairs (bp)) bound more efficiently than shorter DNA duplexes (15 bp) (*Figure 1C*, *Figure 1—figure supplement 2*). The ability of the Scc3T-Scc1K subcomplex to bind DNA depended on the presence of Scc1K. Conversely, the other HEAT-repeat-kleisin subcomplex of cohesin, Pds5-Scc1, the Smc1hd-CScc1 subcomplex and the Wapl subunit did not interact with DNA in this assay. Thus, as in the paralogous condensin complex, the HEAT-repeat protein bound to the C-terminal region of the kleisin subunit directly engages DNA (*Kschonsak et al., 2017*).

To identify the molecular basis of this interaction, we crystallized the Scc3T-Scc1K complex from budding yeast bound to 19 bp of dsDNA. Optimized crystals diffracted anisotropically to a minimum Bragg spacing of 3.6 Å in the best, and ~5.7 Å in the worst direction (*Table 1*). We determined the

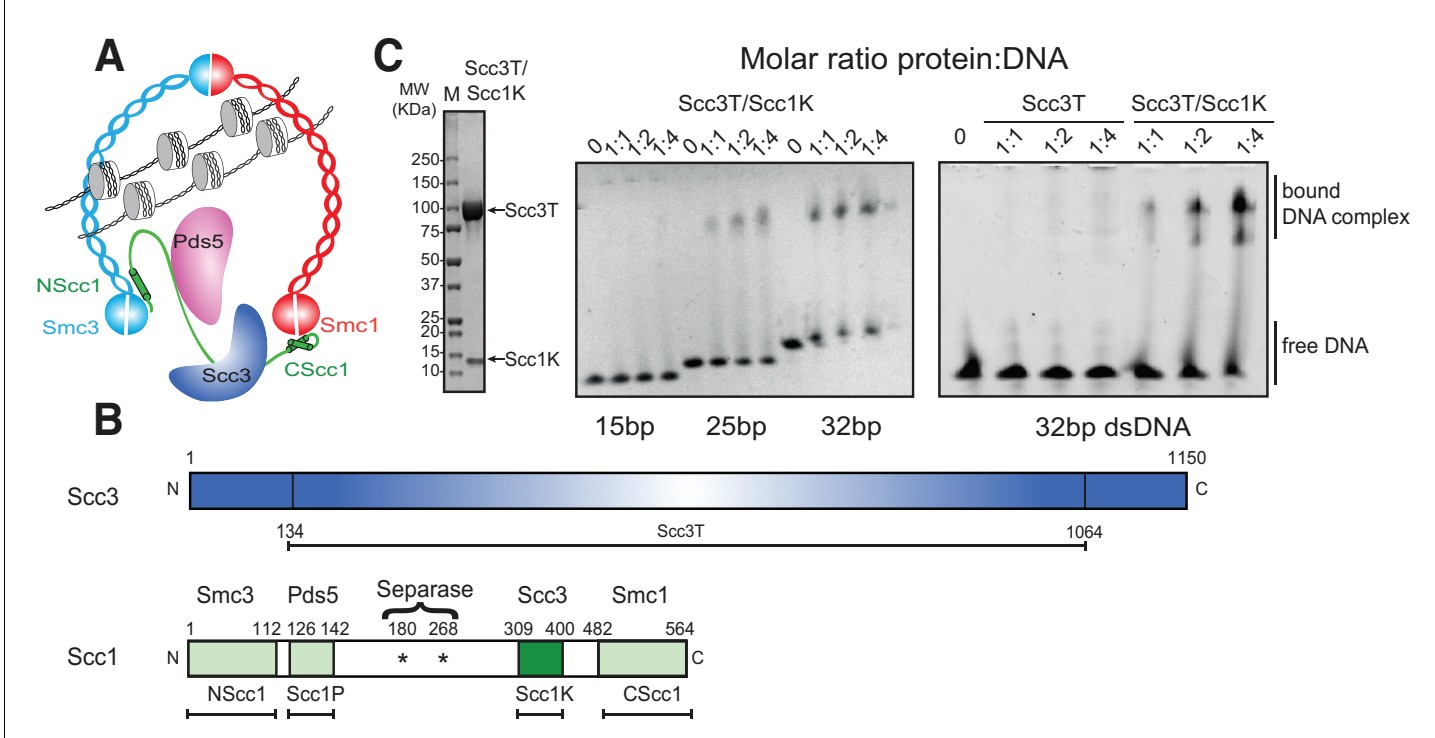

**Figure 1.** DNA binding by the Scc3-Scc1 subcomplex. (**A**) Cartoon of the cohesin complex. (**B**) Domain structure of Scc3 and Scc1. Construct boundaries used and their acronyms are shown below. (**C**) SDS-PAGE analysis of purified Scc3T-Scc1K and DNA binding analysis by EMSA. Scc3 binds to longer DNA more efficiently compared to shorter DNA. The DNA binding capacity of Scc3T is enhanced by Scc1K.

DOI: https://doi.org/10.7554/eLife.38356.002

The following figure supplements are available for figure 1:

**Figure supplement 1.** Cartoon of domain boundaries and constructs used.

DOI: https://doi.org/10.7554/eLife.38356.003

**Figure supplement 2.** SDS-PAGE analysis of the purified cohesin components and DNA binding analysis by EMSA.

DOI: https://doi.org/10.7554/eLife.38356.004

structure by molecular replacement, using the structures of an Scc3 ortholog from *Zygosaccharomyces rouxii* and a C-terminal fragment of *Saccharomyces cerevisiae* Scc3 as search models (**Roig et al., 2014**). The resulting electron-density map provided a continuous trace of the polypeptide main chain, but with a limited level of detail owing to the anisotropy of the data. Despite these drawbacks, we successfully traced the structure using a selenomethionine derivative and refined a model encompassing amino acid residues 134 to 1064 of Scc3 in complex with residues 309 to 400 of Scc1, bound to a 19 bp DNA molecule (**Table 1**).

As seen in other structures of Scc3, the protein is hook-shaped in the C-terminal section and contains an N-terminal 'nose' formed by a pair of extended antiparallel α-helices (**Figure 2A**). Similarly to the interaction of human Scc1 with Scc3, the yeast Scc1 in our structure binds along the convex surface of the hook-shaped HEAT-repeat subunit. We detected additional electron density corresponding to dsDNA within the cradle of this hook (**Figure 2—figure supplement 1A,B**). Whereas the DNA duplexes aligned to form pseudocontinuous double helices throughout the crystal, the DNA duplex was slightly too short for tight end-to-end stacking (**Figure 2—figure supplement 1A, B**). As a result, the DNA density was only partially resolved, apparently due to rotational and translational disorder of the DNA in the binding cavity.

To identify amino acid residues potentially involved in DNA binding, we mapped the electrostatic surface potential onto the Scc3-Scc1 structure (**Figure 2B**). This revealed that DNA is nested within an extended cradle spanning the majority of Scc3-Scc1, lined by a set of positively charged residues that directly contact the DNA phosphate backbone. The DNA is aligned almost parallel to the N-terminal 'nose' of Scc3, which interacts through a set of positively charged amino acid residues with the

**Table 1.** Data collection and refinement statistics.

| | Scc3T/Scc1K native | Scc3T/Scc1K SeMet |
|---|---|---|
| Data collection | | |
| Space group | $P2_12_12$ | $P2_12_12$ |
| Cell dimensions | | |
| $a$, $b$, $c$ (Å) | 109.9, 115.4, 295.6 | 109.9, 115.6, 296.2 |
| Wavelength (Å) | 1.282 | 1.282 |
| Resolution (Å) | 50–3.60 | 49.9–4.79 |
| No. reflections | 20963 | 10279 |
| $R_{merge}$ | 5.8 (122.6)* | 4.6 (112.3)* |
| $I / \sigma I$ | 11.9 (1.6)* | 10.6 2(.1) |
| $CC\ 1/2$ | 0.99 (0.56) | 0.99 (0.52) |
| Completeness (%) | 91.4 (63.5)* | 93.6 (71.2)* |
| Redundancy | 4.4 (6.0)* | 1.8 (1.8) |
| Refinement | | |
| Resolution (Å) | 50–3.60 | |
| $R_{work}/R_{free}$ | 0.28/0.31 | |
| No. atoms | 16465 | |
| Protein | 14909 | |
| DNA | 1556 | |
| $B$-factors (mean) | | |
| Protein | 254.5 | |
| DNA | 266.4 | |
| R.m.s deviations | | |
| Bond lengths (Å) | 0.002 | |
| Bond angles (°) | 0.5 | |

*Values in parentheses are for highest-resolution shell.
DOI: https://doi.org/10.7554/eLife.38356.005

DNA of a neighbouring complex related by crystallographic symmetry (*Figure 2—figure supplement 1C*). We observed no direct nucleotide base–amino acid interactions, which explains the apparent lack of DNA sequence specificity.

To ascertain the amino-acid register of Scc1, we used an anomalous difference map peak for M373 in the selenomethionine-derivative data (*Figure 2—figure supplement 1D*). The deduced register places Scc1 residue K363 in close proximity to the DNA (*Figure 2A*, *Figure 2—figure supplement 1D*). Direct interactions between Scc1 and the DNA phosphate backbone potentially explain why the Scc3T-Scc1K subcomplex has greater DNA binding affinity than does isolated Scc3T (*Figure 1C*).

Mapping of sequence conservation onto the structure revealed that amino acid residues in the DNA binding groove are generally well conserved among yeast Scc3 orthologs (*Figure 3A*). In particular, amino acid residues that are located proximal to the DNA phosphate backbone showed strong conservation. To further evaluate the contributions made by individual segments of the DNA-binding surface, we subdivided participating residues into a series of three patches, based on their physical proximity to the DNA (*Figure 3B*), and subjected these patches to site-directed mutagenesis.

We measured the DNA equilibrium dissociation constant by fluorescence polarization, using a 32-bp 6-FAM labelled DNA substrate, which is sufficient to bridge the entire DNA binding surface present in the Scc3-Scc1 crystal structure (*Figure 3C*). Whereas the wild-type Scc3T-Scc1K complex bound this substrate with an equilibrium dissociation constant of 2.2 µM, charge-inversion mutations

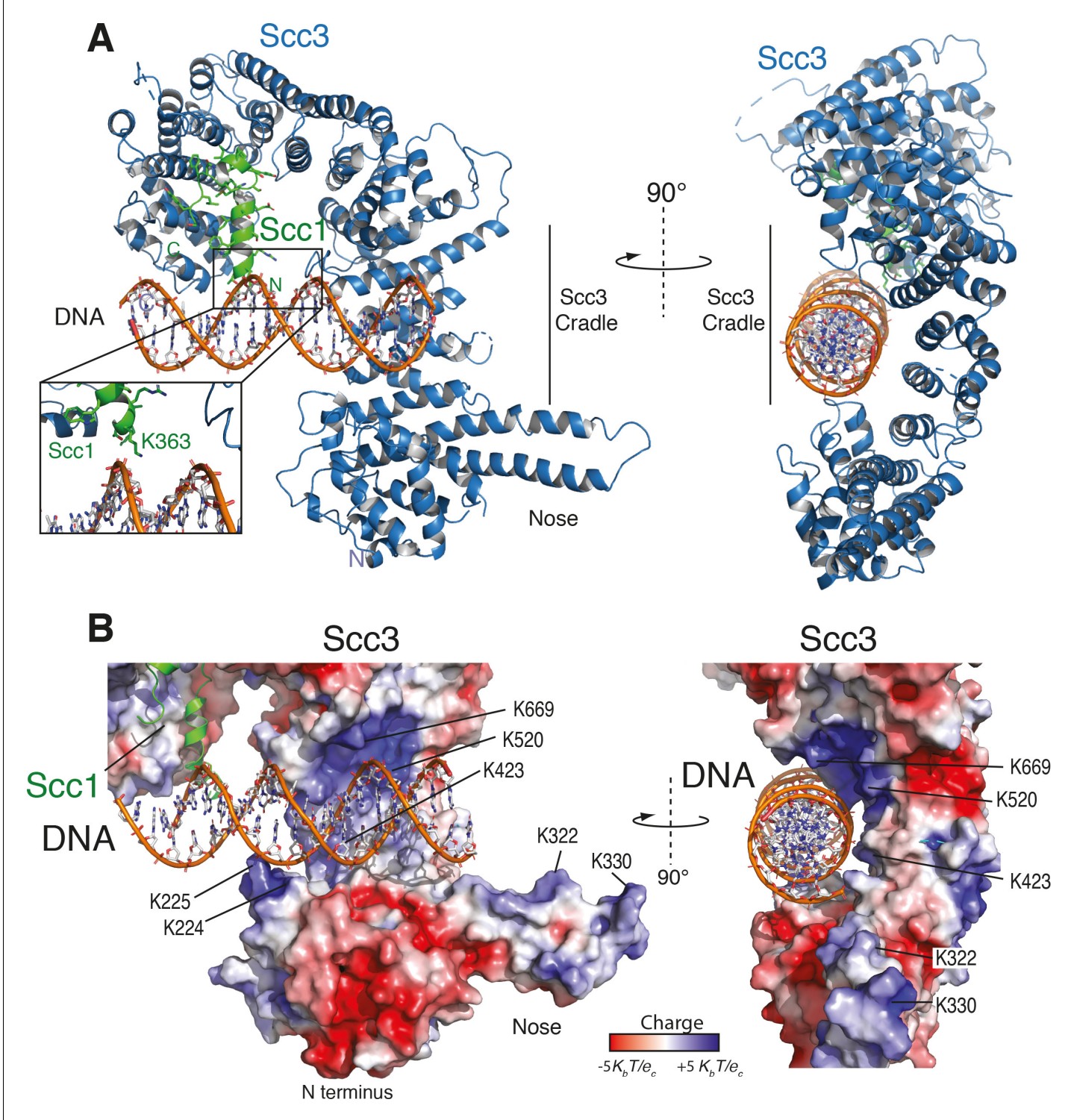

**Figure 2.** Structure of the Scc3-Scc1 subcomplex bound to DNA. (**A**) Cartoon representation of the Scc3-Scc1 complex bound to a 19 bp dsDNA substrate. The N- and C- termini of Scc3 (violet) and Scc1 subunits (green) are shown. The inset shows a close-up view of the Scc1 amino acid K363. (**B**) Electrostatic surface potential representation of the Scc3-Scc1 subcomplex with bound dsDNA (calculated with APBS and displayed with Pymol).

DOI: https://doi.org/10.7554/eLife.38356.006

The following figure supplement is available for figure 2:

**Figure supplement 1.** Electron density for the DNA molecule bound to the Scc3-Scc1 subcomplex.

DOI: https://doi.org/10.7554/eLife.38356.007

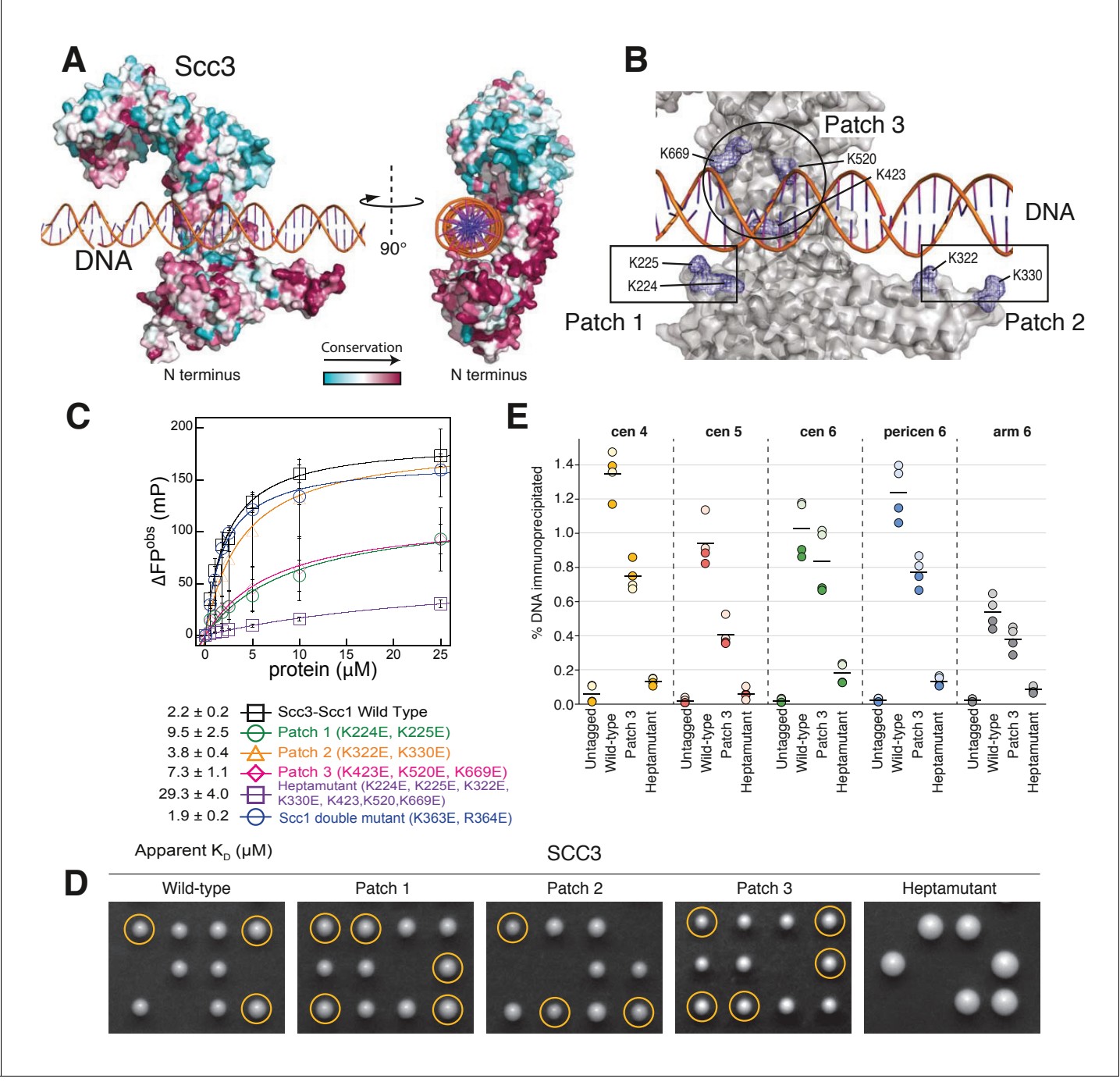

**Figure 3.** A conserved DNA binding domain in the Scc3-Scc1 subcomplex is required for cohesin association with chromatin. (A) Surface amino acid conservation of yeast Scc3. Residues in the DNA binding domain are well conserved. (B) DNA binding residues are located in three surface patches of Scc3. (C) DNA binding fluorescence polarization of 6-FAM labelled 32 bp dsDNA by variants of the Scc3-Scc1 subcomplex. Data points corresponding to the average of three independent experiments were fitted to a standard binding equation assuming a single binding site using Kaleidagraph. Standard deviations are depicted as vertical error bars. Apparent dissociation constants ($K_D$) are noted below. (D) Tetrad analysis of diploid budding yeast strains expressing ectopic wild-type or mutant versions of Scc3 under control of the endogenous promoter in an *SCC3/scc3Δ* background (strains C5013, C5014, C5015, C5043, C5033). Images were recorded after three days at 30°C on rich media. Genetic marker analysis identified *Scc3(mutant)*, *scc3Δ* cells (circles). (E) ChIP-qPCR analysis of binding to centromeric (cen), pericentromeric (pericen) or chromosome arm (arm) regions (chromosomes IV, V, and VI as indicated) of untagged (strain C3) or $PK_6$-tagged wild-type or mutant versions of Scc3 expressed from an ectopic locus under its endogenous promoter (strains C5013, C5043, C5033). The fractions of immunoprecipitated DNA relative to input DNA are plotted as circles for two biological repeats with two technical repeats each (same colour pairs). Mean values of all four data points are shown as lines.

*Figure 3 continued on next page*

*Figure 3 continued*

DOI: https://doi.org/10.7554/eLife.38356.008

The following figure supplements are available for figure 3:

**Figure supplement 1.** Biochemical analysis of Scc3-Scc1 subcomplexes and Scc3 conservation alignment.
DOI: https://doi.org/10.7554/eLife.38356.009

**Figure supplement 2.** In vivo analysis of SCC3 mutants.
DOI: https://doi.org/10.7554/eLife.38356.010

in patches 1 and 3, located at the crest of and within the Scc3 cradle, had only modest effects on affinity (equilibrium dissociation constants of 9.5 μM and 7.3 μM, respectively). Patch 2 mutants, residing in the Scc3 'nose', exhibited essentially unaltered DNA binding affinity. In contrast, the simultaneous mutation of all three patches (a heptamutant) reduced the binding affinity of the patch 1 or 3 mutants even further (to 29.3 μM, *Figure 3C*). The defect in binding DNA was not due to any impact of the mutations on the structural integrity of Scc3-Scc1, as all mutant complexes eluted indistinguishably from wild-type Scc3-Scc1 during size-exclusion chromatography (*Figure 3—figure supplement 1A*) and efficiently formed a complex with Scc1 in vitro (*Figure 3—figure supplement 1B*) or in vivo (see below). We conclude that the positively charged Scc3 cradle comprises the major DNA-binding surface of Scc3 and that the positively charged amino acid residues located in patch 1 and 3 constitute a composite DNA binding surface. The distribution of these residues across an extended surface of Scc3 might explain why their significance has thus far eluded cell-biological and genetic characterization. To test whether the direct interactions between Scc1 and the DNA phosphate backbone contribute to DNA binding we introduced the double mutation K363E/R364E into Scc1K. This mutant showed an essentially indistinguishable equilibrium dissociation constant as compared to that of the wild-type complex, thus indicating that this positively charged patch of Scc1 does not contribute to DNA binding (*Figure 3C*).

We then assayed whether DNA binding by Scc3 is important for cohesin function in vivo. We integrated wild-type or mutant versions of Scc3 into a diploid yeast strain in which one of the two endogenous *SCC3* genes had been deleted (*Supplementary file 2*) and analysed the competence of these Scc3 variants to complement the *SCC3* deletion by tetrad dissection (*Figure 3D*). Whereas all three individual patch mutants could support cell proliferation, cells expressing the heptamutant mutant version as their only source of Scc3 failed to divide, despite expressing the mutant Scc3 protein at wild-type levels (*Figure 3—figure supplement 2A*).

To test whether the inability of the heptamutant version of Scc3 to support cohesin function was the result of a defect in the association of the mutant cohesin complex with chromosomes, we measured the levels of wild-type and mutant cohesin complexes at five independent binding sites in the budding yeast genome by chromatin immunoprecipitation followed by quantitative PCR (ChIP-qPCR). The amounts of chromosomal DNA that co-immunoprecipitated with the Scc3 patch 3 mutant were on average 40% lower than those that co-immunoprecipitated with wild-type Scc3. The Scc3 heptamutant failed to bind chromatin entirely (*Figure 3E*), although it was incorporated normally into cohesin complexes (*Figure 3—figure supplement 2B*). We conclude that DNA binding by Scc3-Scc1 is essential for the stable association of cohesin complexes with chromosomes in vivo and hence an important determinant of cohesin function.

## Discussion

Targeting of cohesin to the genome is essential for numerous aspects of chromosome biology, including sister chromatid cohesion, DNA damage repair, and transcriptional regulation (*Uhlmann, 2016*). In this study, we identified a direct DNA-binding site in the Scc3 subunit of the cohesin complex and determined its interaction with DNA at near-atomic resolution. These findings provide evidence for a site of direct DNA contact in cohesin complexes, which presumably contributes to the initial step of chromosome entrapment and/or DNA translocation (*Figure 4C*).

We show here that cohesin interacts directly with DNA through a complex formed by the HEAT-repeat subunit Scc3 and the kleisin subunit Scc1. We have determined the DNA-bound structure of Scc3 in complex with a minimal fragment of Scc1 and demonstrated that DNA binding depends on conserved positively charged residues of a composite Scc3-Scc1 interface. We have used this

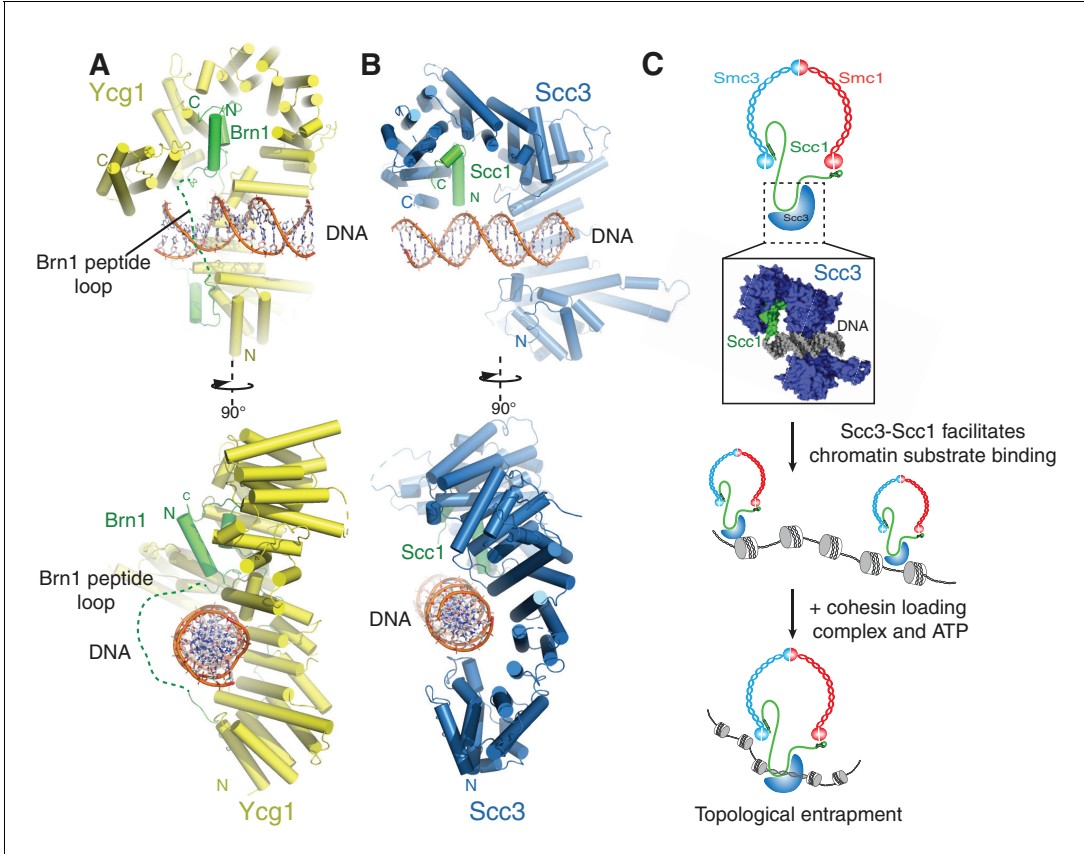

**Figure 4.** A conserved DNA binding interface in cohesin and condensin. (**A**) Structure of the DNA-bound Ycg1–Brn1 subcomplex from condensin. (**B**) Structure of the Scc3-Scc1 complex. In the condensin structure, a peptide loop (green dashed line) of the Brn1 kleisin subunit encircles the bound DNA and prevents its dissociation. Alignments were generated by secondary structure matching using Cα atoms from the Scc3 HEAT-repeats and the structurally equivalent region of the condensin Ycg1 HEAT-repeat subunit. (**C**) Model for Scc3-mediated DNA binding by cohesin complexes. Scc3-Scc1 enables direct chromatin binding. Cohesin is loaded by Scc2-Scc4 in an ATP dependent fashion thus resulting in topological DNA entrapment.
DOI: https://doi.org/10.7554/eLife.38356.011

structure to derive DNA-binding deficient Scc3 variants, which fail to support cohesin recruitment to chromatin and consequently cell division. In addition to providing a scaffold for the assembly of cohesin regulators (*Hara et al., 2014*; *Murayama and Uhlmann, 2014*; *Roig et al., 2014*) and thereby participating in the maintenance of cohesion, the Scc3-Scc1 subcomplex also plays a key role in DNA substrate recognition and hence the efficient association of the cohesin holocomplex with chromatin.

The cohesin ring has been proposed to topologically embrace chromosomal DNA, which requires that DNA is loaded into and also released from the ring complex during cohesin's reaction cycle (*Nasmyth, 2011*). As mutations in Scc3 that prevent DNA engagement by the Scc3-Scc1 subcomplex in vitro fail to stably bind to chromatin in vivo, it is likely that direct DNA-cohesin interactions contribute to such DNA entry and exit reactions (*Figure 4C*).

We propose that the Scc3-Scc1 subcomplex provides a dynamic DNA anchoring point that is required for the efficient loading and/or maintenance of cohesin on chromatin. Such a model for Scc3 is supported by previous data, which indicate that Scc3 contributes to cohesin loading (*Hu et al., 2011*; *Murayama and Uhlmann, 2014*; *Orgil et al., 2015*; *Roig et al., 2014*). In agreement with this model, Scc1 deletion mutants that lack the sequence responsible for binding Scc3 fail to load onto yeast chromosomes (*Hu et al., 2011*). Furthermore, Scc3 enhances topological DNA entrapment by cohesin in in vitro loading assays (*Murayama and Uhlmann, 2014*). Thus, DNA binding by the Scc3-Scc1 subcomplex might be the first step in moving chromosomal DNA into the

cohesin ring. The subsequent entrapment reaction is then presumably catalysed by the Scc2-Scc4 complex (*Davidson et al., 2016*; *Murayama and Uhlmann, 2014*; *Stigler et al., 2016*).

Surface patch 3 mutations in Scc3, which partially ablate DNA binding by the Scc3-Scc1 subcomplex and reduce cohesin levels on chromatin, do not exhibit any obvious growth defects and are therefore presumably competent to establish sister chromatid cohesion. Indeed, partial depletion of cohesin does not seem to impact some of its core functions, including sister-chromatid cohesion and chromosome segregation (*Elbatsh et al., 2016*; *Heidinger-Pauli et al., 2010*). In contrast, even mutations that only slightly impair DNA binding of the equivalent HEAT-repeat/kleisin module of the condensin complex are sufficient to abolish stable chromatin association and to interfere with cell division (*Kschonsak et al., 2017*). Such discrepancies might be due to alternate loading and/or maintenance mechanisms of cohesin and condensin. Whilst cohesin is loaded by the Scc2-Scc4 complex, no such independent loading factor has been identified for condensin thus far, which could explain why the latter depends more strongly on the direct DNA binding site formed by its HEAT-repeat and kleisin subunits.

Binding of condensin to DNA is further stabilized by the entrapment of the bound DNA helix within a kleisin peptide loop (*Figure 4A*-) (*Kschonsak et al., 2017*). The relevant section of Scc1 that would contribute to such topological DNA entrapment is disordered in our structure(*Figure 4B*). As Scc1 is clearly required for DNA binding of the Scc3-Scc1 subcomplex, but apparently not through direct DNA interactions (*Figure 3C*), it is possible that cohesin uses a similar mode of chromatin engagement. These findings thus point towards a conserved molecular mechanism that enables chromatin substrate engagement by condensin and cohesin. This mechanism potentially facilitates topological loading, chromatin looping and tracking along chromatin fibres by these SMC complexes (*Ganji et al., 2018*).

## Materials and methods

### Constructs, expression and purification

Scc3, Scc1 and other cohesin subunits were amplified from yeast Genomic DNA (Millipore). Scc3 and mutant variants thereof were cloned into pETM-30 vector using *NcoI* and *NotI* restriction enzyme cleavage sites. Scc1 constructs were cloned using the *NcoI* and *NotI* sites of a pACYC-DUET vector for co-expression with Scc3, or into vectors containing the cognate Smc head domain in their second ORF for Smc/kleisin complexes (see below). Codon optimised genes comprising the Smc1 and Smc3 ATPase domains were produced by gene synthesis (Thermofisher), to include C-terminal 6xHistidine tags, and ligated into the *NdeI-XhoI* sites of pRsf-DUET1 and pACYC-DUET1 respectively via Gibson Assembly (NEB). Pds5 and Wapl were cloned and expressed as described previously (*Muir et al., 2016*). For the expression and purification of Pds5, media and buffers were supplemented with 20 μM and 5 μM of inositol hexa-kis-phosphate, respectively (*Ouyang et al., 2016*).

Proteins were expressed in *Escherichia coli* BL21(*DE3*) by auto-induction (*Studier, 2005*). Cells were grown at 37°C until $OD_{600nm} = 0.6$ and then shifted to 18°C for 16 hr. Cells were harvested and washed once with ice-cold PBS buffer. Pellets were resuspended in buffer 1 (40 mM TRIS, pH 7.5, 500 mM NaCl, 20 mM imidazole, 0.5 mM TCEP) containing one tablet of complete, Mini, EDTA-free protease inhibitors (Roche). Cells were lysed using a microfluidiser (Microfluidics) and centrifuged at 15000 rpm for 1 hr using a JA-20 rotor (Beckman). The supernatant was loaded onto 5 ml $Co^{2+}$–conjugated IMAC beads (GE healthcare) by using a peristaltic pump (GILSON). The column was washed with 10 column volumes of buffer 1 and the protein eluted with buffer 2 (40 mM TRIS, pH 7.5, 300 mM NaCl, 300 mM imidazole, 0.5 mM TCEP). The His-GST tag was cleaved by addition of His-tagged TEV protease during overnight dialysis against 40 mM TRIS, pH 7.5, 300 mM NaCl, 0.5 mM TCEP at 4°C. For Smc3-NScc1, Smc1-CScc1, and the Smc3-Smc1 hinge, this cleavage step was bypassed.

The HIS-GST tag, protease and uncleaved protein were removed by passing this mixture over $Co^{2+}$ IMAC resin. The flow through was concentrated using an Amicon Ultra −15 concentrator (Millipore) and applied onto a MonoQ 5/50 GL column (GE healthcare) in buffer 3 (150 mM NaCl, 40 mM TRIS, pH7.5, 0.5 mM TCEP). Proteins were eluted using a linear gradient in buffer 4 (1 M NaCl, 40

mM TRIS, pH 7.5, 0.5 mM TCEP). The final purification step was performed by using a HiLoad 16/60 Superdex 200 prep–grade column in buffer 5 (150 mM NaCl, 20 mM TRIS, pH7.5, 0.5 mM TCEP).

## DNA binding assays

For analysis of DNA-binding by EMSA and for co-crystallization, DNA substrates were generated by annealing complementary oligonucleotides (MWG Eurofins) at a final concentration of 1 mM in 20 mM TRIS, pH 7.5, 150 mM NaCl (*Supplementary file 1*). Successful annealing and purity of the oligonucleotides were confirmed by native PAGE on a 6% gel.

For EMSA experiments, varying concentrations of protein samples were incubated at the indicated molar ratios with 1 µM the 32mer DNA in 150 mM NaCl, 20 mM TRIS, pH 7.5, 0.5 mM TCEP. Samples were incubated on ice for 30 min. 5% Glycerol was added and the samples were analysed on a 6% native 1x TRIS-Glycine (25 mM TRIS, 250 mM glycine, pH 8.3, 5% Glycerol) polyacrylamide gel using 1x TRIS-Glycine running buffer. Gels were stained with SYBR Safe (Thermo Fisher Scientific) to visualize DNA-bound complexes or Coomassie Blue for protein staining.

## Fluorescence Polarization (FP)

32 bp 6-FAM labelled DNA was prepared by annealing two complementary DNA strands, essentially as described for the crystallisation oligonucleotides, albeit under low-light conditions (*Supplementary file 1*). Fluoresence polarisation assays were conducted in a buffer containing 50 mM TRIS pH 7.5, 150 mM NaCl, 0.1% tween and 0.5 mM TCEP. A series of protein concentrations, ranging from 0.5 µM to 25 µM, were incubated in the presence of 50 nM DNA for 30 min at room temperature in order to attain equilibrium. Immediately thereafter, fluorescence polarization was recorded using 485 nm and 520 nm excitation and emission filters, respectively (CLARIOstar, BMG Labtech, Germany). The change in fluorescence polarization was then plotted as mean values of three independent replicates and the dissociation constant for each complex determined.

## Crystallization and data collection

Crystals of the Scc3-Scc1-DNA complex were obtained by mixing 8 mg ml$^{-1}$ protein with DNA at a 1:1.1 ratio. 1 µl of the protein:DNA complex were mixed with 1 µl 10% PEG 8000, 0.1M Bis-TRIS, pH 6.5 crystallization buffer and equilibrated against the crystallization buffer at 4°C. Initial crystals with a 17 mer DNA were obtained after 5 days. These crystals were used as seeds for crystallization of Scc3-Scc1 bound to a 19mer DNA using the same crystallization condition. Crystals were cryo protected by adding 20% Glycerol to the crystallization buffer and flash frozen in liquid nitrogen. Diffraction data for native and selenomethionine-derivatised Scc3-Scc1-DNA crystals were collected at 100 K at an X-ray wavelength of 0.966 Å at beamline ID30A-1/MASSIF-1 (*Bowler et al., 2015*) of the European Synchrotron Radiation Facility, with a Pilatus3 2M detector using automatic protocols for the location and optimal centring of crystals (*Svensson et al., 2015*). The beam diameter was selected automatically to match the crystal volume of highest homogeneous quality (*Svensson et al., 2018*). Data were processed with XDS (*Kabsch, 2010*) and imported into CCP4 format using AIMLESS (*Winn et al., 2011*).

## Structure determination refinement and analysis

The structure of the Scc3-Scc1-DNA complex was determined by molecular replacement by Phaser (*McCoy et al., 2007*) using a Scc3 structure from *Zygosaccharomyces rouxii* (PDB code 4UVK) and a structure of a C-terminal fragment of Scc3 from *Saccharomyces cerevisiae* (PDB code 4UVL) (*Roig et al., 2014*) in spacegroup P2$_1$2$_1$2 at a resolution of 3.6 Å (*Table 1*). The model was build by iterative rounds of manual adjustments with Coot and of restrained refinements with Phenix (*Afonine et al., 2012*; *Emsley et al., 2010*). Sequence register was confirmed using a selenomethionine-derivatized crystal. Analysis of the refined structure in MolProbity showed that there were no residues in the disallowed and 94% in the favoured region of the Ramachandran plot. The MolProbity all atom clash score was 4.1 (*Chen et al., 2010*). Structures were visualized with PyMOL (Schrödinger, LLC). Surface conservation graphics were created using the ConSurf server (*Ashkenazy et al., 2016*) using a multi-sequence alignment containing Scc3 orthologs from *Saccharomyces_cerevisiae* (P40541), *Saccharomyces_kudriavzevii* (J5PHP0), *Candida glabrata* (A0A0W0DN34), *Naumovozyma castellii* (G0VFI2), *Vanderwaltozyma polyspora* (A7TNN6),

*Tetrapisispora phaffii* (G8BRB9), *Saccharomyces kudriavzevii* (J5PHP0), *Zygosaccharomyces rouxii* (A0A1Q3A0R6), *Torulaspora delbrueckii* (G8ZZR1), *Kluyveromyces lactis* (Q6CIC3), *Lachancea quebecensis* (A0A0P1KU08), *Eremothecium cymbalariae* (G8JTR2), *Ashbya gossypii* (Q75AL6), *Pichia sorbitophila* (G8YM42) and *Yarrowia lipolytica*(Q6C144). The electrostatic surface potential graph was created with APBS (*Baker et al., 2001*).

### Chromatin immunoprecipitation and ChIP-qPCR

Chromatin immunoprecipitation followed by quantitative PCR ChIP-qPCR was performed from asynchronous yeast cell cultures as described (*Cuylen et al., 2011*), with the exception that sonication was performed with a Bioruptor Plus (Diagenode) at 4°C using 6 cycles of 30 s on, 60 s off at 'high' level. Quantitative PCR was performed with primers listed *Supplementary file 3*.

### Tetrad analysis

Wild-type or mutant alleles of *Scc3* fused to a C-terminal $PK_6$ epitope tag were integrated into the *ura3* locus of a *SCC3/scc3::HIS3* heterozygous diploid yeast strain (C1073). Correct integration was confirmed by PCR (*Supplementary file 2*). Following sporulation, strains were tetrad dissected and cultured on YPAD media for 3 days at 30°C before genotyping by replica plating.

### Data availability

Coordinates for the Scc3-Scc1-DNA complex are available from the Protein Data Bank under accession number 6H8Q.

## Acknowledgments

This work was funded by EMBL. We thank Stefan Reich for assistance with FP experiments.

## Additional information

### Funding

| Funder | Author |
|---|---|
| European Molecular Biology Laboratory | Yan Li<br>Kyle W Muir<br>Matthew W Bowler<br>Jutta Metz<br>Christian H Haering<br>Daniel Panne |

The funders had no role in study design, data collection and interpretation, or the decision to submit the work for publication.

### Author contributions

Yan Li, Conceptualization, Data curation, Formal analysis, Investigation, Visualization, Methodology, Writing—review and editing; Kyle W Muir, Conceptualization, Formal analysis, Supervision, Investigation, Visualization, Methodology, Writing—original draft, Project administration, Writing—review and editing; Matthew W Bowler, Methodology; Jutta Metz, Data curation, Methodology; Christian H Haering, Formal analysis, Supervision, Visualization, Methodology, Writing—review and editing; Daniel Panne, Conceptualization, Formal analysis, Supervision, Funding acquisition, Visualization, Methodology, Writing—original draft, Project administration, Writing—review and editing

### Author ORCIDs

Yan Li http://orcid.org/0000-0001-8190-1633
Kyle W Muir http://orcid.org/0000-0002-0294-5679
Matthew W Bowler http://orcid.org/0000-0003-0465-3351
Christian H Haering http://orcid.org/0000-0001-8301-1722
Daniel Panne http://orcid.org/0000-0001-9158-5507

Decision letter and Author response
Decision letter https://doi.org/10.7554/eLife.38356.021
Author response https://doi.org/10.7554/eLife.38356.022

# Additional files

## Supplementary files

• Transparent reporting form
DOI: https://doi.org/10.7554/eLife.38356.012

• Supplementary file 1 DNA substrates
DOI: https://doi.org/10.7554/eLife.38356.013

• Supplementary file 2 Yeast genotypes. All strains are derivatives of W303.
DOI: https://doi.org/10.7554/eLife.38356.014

• Supplementary file 3 ChIP-qPCR primer sequences (5'→3').
DOI: https://doi.org/10.7554/eLife.38356.015

## Data availability

Diffraction data have been deposited in PDB under the accession code 6H8Q.

The following dataset was generated:

| Author(s) | Year | Dataset title | Dataset URL | Database, license, and accessibility information |
|---|---|---|---|---|
| Li Y, Muir KW, Bowler MW, Metz J, Haering CH, Panne D | 2018 | Diffraction data for the Scc3-Scc1-DNA complex | https://www.rcsb.org/structure/6H8Q | Publicly available at the RCSB Protein Data Bank (accession no. 6H8Q) |

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
