## [Decision Letter]

Thank you for submitting your article "Structural basis for Scc3-dependent cohesin recruitment to chromatin" for consideration by *eLife*. Your article has been reviewed by Andrea Musacchio as the Senior Editor, a Reviewing Editor, and three reviewers.. The reviewers have opted to remain anonymous.

The reviewers have discussed the reviews with one another and the Reviewing Editor has drafted this decision to help you prepare a revised submission. We hope you will be able to submit the revised version within two months.

Summary:

Li et al., describe a crystal structure of Scc3 (slightly truncated at both ends and designated Scc3T) with a fragment of Scc1 ("Scc1K") bound with a 19bp fragment of dsDNA. The structure, which closely resembles a corresponding condensin DNA complex published last year by Haering's group, shows that DNA binds along a groove created by the curvature of the Scc3 heat repeats. Mutations of residues at points of contact with DNA backbone diminish affinity in vitro and (in appropriate combination) prevent loading in vivo, as shown by ChIP-qPCR and by tetrad analysis of strains ectopically expressing wt or mutant versions of Scc3 in an *SCC3/scc3Δ* background. The biochemical and structural work clearly identify the molecular basis of an interaction between Scc3 and DNA and demonstrate that a fragment of Scc1 enhances this interaction. While not entirely unsurprising, the results add to our understanding of how cohesin might function mechanistically. With appropriate revision, the MS can be made suitable for publication in *eLife*.

Essential revisions:

1) In the Discussion section (first paragraph), the authors set up a straw man. The only alternative to direct protein-DNA contacts (either by a cohesin subunit or by some adaptor protein) is that cohesin is snapping open and closed all the time and should it happen to entrap DNA, it somehow sticks. That silly alternative is obviously hugely improbable. Thus, the real conclusion is that "these findings provide evidence for a site of DNA contact, probably during the initial step in chromosome entrapment" – or something like that. See also point 3, below. Incidentally, "topological principles" don't "drive" anything in biology – they explain various mechanisms or activities and suggest why they may have evolved, but biology is never "driven" by "principles" other than natural selection (except in the minds of the unreconstructed Cartesians still running around in France).

2) The anisotropy of the crystals, presumably due to a less than optimal DNA length, limits the accuracy and the information content of the structure. Did the authors, seeing the result, take the obvious next step of trying to get better crystals with other DNA lengths (e.g., 20 or 21 bp)? (That is, were all DNAs in Supplementary file 2 used in crystallization trials, or just for FP measurements? If the former, why not also 20 bp?) In this reviewer's view, that would have been an easier and better path than the one they took by validating the Scc1 trace with SeMet. In any case, Figure 3—figure supplement 1 should show either the DNA density in the initial MR map or (if the MR is good enough – it might not be) an Fo-Fc map after that initial step (i.e., phases from MR but 2Fo-Fc, showing in principle what's missing). In any case, a map with the final 2Fo-Fc phases, which included this DNA contributions, is not helpful. Incidentally, in Supplementary file 1, the *R*_merge_ in the last bin is truly miserable. Was there an "elliptical" (i.e., anisotropic) cutoff, or did the meaningless reflections in the "bad" directions contribute? If the latter, then please recalculate with the correct anisotropic cutoff for each frame or set of frames, so that pure noise doesn't contribute to the data used. Also, "3.99" is 4.0 in my book, not "3.9".

3) The text overstates some of the biological and mechanistic conclusions. Although the correlation of affinity in vitro with function in vivo does permit the inference that the observed interaction is part of the DNA docking mechanism, the results do not rule out the participation of other contacts. Indeed, were those contacts strengthened by compensating mutations, it is possible that this contact would not be "indispensible", as the Abstract states.

4) The most useful conclusion is the similarity with condensin. For understandable "psychological" reasons, the authors do not mention the Kschonsak et al. (2017) paper in the Introduction. They should do so, as it surely guided their strategy at some point, either consciously or otherwise. Is the peptide loop definitely absent here, or could its absence be a consequence of truncating Scc1?

5) At the end of the Discussion section, the authors write that the mechanism they describe would enable cohesin to entrap a second DNA helix without releasing the first, etc. Not obvious to this reader why or how, perhaps because Figure 4C is so vague and incomprehensible.

6a) Is K363 on Scc1 important for the DNA binding ability of the complex? It would also be useful to highlight this residue on Figure 2. This experiment is required to confirm the in vivo relevance of the enhancement of DNA association by the Scc1 fragment in the in vitro experiments.

6b) Scc1 is clearly required for the DNA-binding activity of Scc3-Scc1. The structure suggests that Scc1 K363 might contact DNA. Does Scc1 K363E reduce the binding of Scc3-Scc1? Related to this, even though the authors cannot see any density of the N-terminal region of Scc1K, this region might contribute to DNA binding. This should be experimentally tested. In the human SA2-Scc1K structure, the corresponding region in Scc1K forms a helix that is located at the base of the "nose" of SA2. Can the authors build a model of the SA2-Scc1-DNA complex and see if this N-terminal region of Scc1K might be close to DNA?

NOTE: Concern on the function of K363 was raised by two reviewers and is reported here in its original wording as points 6a and 6b.

7) The ChIP-qPCR is essential for the conclusions of the paper but there are some issues with the presented experiment in Figure 3D. A minimum of 3 biological repeats are required to compute standard deviation, so the error bars here are not appropriate and should be removed. The authors could show the data for the two biological replicates side by side without error bars as an indication of reproducibility or, better, repeat the experiment a third time and calculate standard deviation. What do the percentages mean above the bars? How did the authors analyse the heptamutant given that it is not viable? Do the cells also carry endogenous Scc3? In this case, do all the strains in this experiment carry untagged Scc3 in addition to the tagged wild type or mutant protein? The fact that patch 2 mutants still bind DNA in vitro predicts that the patch 2 mutant protein should also associate with the chromosome in vivo, but this was not tested.

8) The authors are proposing that the reported interactions are required for cohesin loading. However, an alternative possibility is that "core" cohesion can load but that Scc3 fails to associate with it. This should be tested by assessment of the association of other "core" cohesin subunits (Smc1/Smc3/Scc1) with chromosomes in the patch 3 mutant cells.

9) What is the effect of the observed mutations on sister chromatid cohesion? The authors should test this using the TetR-GFP or LacI-GFP system.

10) Hara et al. (2014) showed that mutating conserved basic residues in the N-terminal and middle regions of Scc1K did not affect Scc1K binding to SA2. These regions may transiently dissociate from SA2/Scc3 while the C-terminal region of Scc1K is still anchored to SA2/Scc3. It is thus possible that Scc3-Scc1K form a topological embrace of DNA, similar to the condensin sub-complex. This possibility needs to be discussed, especially if the N-terminal region of Scc1K is required for DNA binding (see point 6b).

[Editors' note: further revisions were requested prior to acceptance, as described below.]

Thank you for submitting your article "Structural basis for Scc3-dependent cohesin recruitment to chromatin" for consideration by *eLife*.

I have now examined your resubmission and I am happy to inform you that I consider it essentially ready for acceptance. However, before formal acceptance, I would like to note the following three points:

1) You seem to be using two somewhat different color schemes for Scc3 in the different figures, more bluish in Figure 2 and Figure 4, and more violet in Figure 1. May I suggest that you make the colours more uniform?

2) In Figure 2—figure supplement 1E, the right hand panel appears to be a composite of pasted lanes. If this is the case, could you please clearly mark this on the figure with a black vertical line and add a short reference to lane pasting in the legend?

---

## [Author Response]

Essential revisions:1) In the Discussion section (first paragraph), the authors set up a straw man. The only alternative to direct protein-DNA contacts (either by a cohesin subunit or by some adaptor protein) is that cohesin is snapping open and closed all the time and should it happen to entrap DNA, it somehow sticks. That silly alternative is obviously hugely improbable. Thus, the real conclusion is that "these findings provide evidence for a site of DNA contact, probably during the initial step in chromosome entrapment" – or something like that. See also point 3, below. Incidentally, "topological principles" don't "drive" anything in biology – they explain various mechanisms or activities and suggest why they may have evolved, but biology is never "driven" by "principles" other than natural selection (except in the minds of the unreconstructed Cartesians still running around in France).

We agree with the reviewers’ point and revised our statement accordingly (Discussion section et seq.): ‘In this study, we identified a DNA-binding site in the Scc3 subunit of the cohesin complex and determined its interaction with DNA at near-atomic resolution. These findings provide evidence for direct DNA-protein contacts in the cohesin complex, which presumably play a major role in the initial step of chromosome entrapment and/or possibly DNA translocation.’

2) The anisotropy of the crystals, presumably due to a less than optimal DNA length, limits the accuracy and the information content of the structure. Did the authors, seeing the result, take the obvious next step of trying to get better crystals with other DNA lengths (e.g., 20 or 21 bp)? (That is, were all DNAs in Supplementary file 2 used in crystallization trials, or just for FP measurements? If the former, why not also 20 bp?) In this reviewer's view, that would have been an easier and better path than the one they took by validating the Scc1 trace with SeMet.

Improving crystal quality by testing different DNA lengths was indeed our strategy. As indicated in the Materials and methods section, the initial DNA duplex length for spontaneous crystallization was 17 bp. We have tried extensively to improve these crystals using DNA of various lengths, including those shown in Supplementary file 1 (previously Supplementary file 2). We obtained the best-diffracting crystals by using the 17-bp DNA crystals as ‘seeds’ for crystallization of a complex containing a 19mer DNA. Our expectation was that a DNA substrate of 20/21bp would be ideal, but none of the longer DNA duplexes allowed crystallization.

In any case, Figure 3—figure supplement 3 should show either the DNA density in the initial MR map or (if the MR is good enough – it might not be) an Fo-Fc map after that initial step (i.e., phases from MR but 2Fo-Fc, showing in principle what's missing). In any case, a map with the final 2Fo-Fc phases, which included this DNA contributions, is not helpful. Incidentally, in Supplementary file 1, the R_merge_ in the last bin is truly miserable. Was there an "elliptical" (i.e., anisotropic) cutoff, or did the meaningless reflections in the "bad" directions contribute? If the latter, then please recalculate with the correct anisotropic cutoff for each frame or set of frames, so that pure noise doesn't contribute to the data used. Also, "3.99" is 4.0 in my book, not "3.9".

We now show a 2Fo-Fc map after the initial MR step, in which the DNA was omitted from the model (Figure 3—figure supplement 1A).

We thank the reviewer for pointing out the high R_merge_ in the highest resolution bin. Indeed, we had not applied an elliptical cutoff and noisy reflections contributed to the data used. We have reprocessed the data using an elliptical cutoff, as suggested. Overall statistics improved and we were now able to include reflections up to 3.6Å. We have refined the model (maintaining and extending the previous set of R_free_ reflections) against this reprocessed data and have updated Table 1 accordingly.

To ensure that potential users have access to the data prior to additional processing, we have deposited unprocessed structure factors. Table 1 now shows data statistics after additional anisotropic data processing.

3) The text overstates some of the biological and mechanistic conclusions. Although the correlation of affinity in vitro with function in vivo does permit the inference that the observed interaction is part of the DNA docking mechanism, the results do not rule out the participation of other contacts. Indeed, were those contacts strengthened by compensating mutations, it is possible that this contact would not be "indispensible", as the Abstract states.

We have revised our Abstract to avoid any overstatement: ‘These findings suggest that the Scc3-Scc1 DNA-binding interface plays a central role in the recruitment of cohesin complexes to chromosomes and therefore for cohesin to faithfully execute its functions during cell division.’

4) The most useful conclusion is the similarity with condensin. For understandable "psychological" reasons, the authors do not mention the Kschonsak et al. (2017) paper in the Introduction. They should do so, as it surely guided their strategy at some point, either consciously or otherwise. Is the peptide loop definitely absent here, or could its absence be a consequence of truncating Scc1?

We now refer to the paper by Kschonsak et al. (2017) in the Introduction: ‘Recent work identified a direct DNA-binding site in the paraloguous condensin complex, where the HEAT repeat subunit Ycg1 in complex with the kleisin subunit Brn1 contact the DNA double helix backbone and stabilize its association through DNA entrapment within a Brn1 peptide loop (Kschonsak et al., 2017).’

We cannot exclude for certain that the peptide loop is lacking in Scc1, as it might be absent due the truncation construct used in our experiments. However, as mentioned previously, data from the Nasymth lab (Chan et al., 2012; Roig et al., 2014; Petela et al., 2018) indicate that Scc1 constructs with mutations/truncations in the relevant area are viable, thus indicating that such a peptide loop, if present at all, may not be essential for cohesin function in budding yeast.

5) At the end of the Discussion section, the authors write that the mechanism they describe would enable cohesin to entrap a second DNA helix without releasing the first, etc. Not obvious to this reader why or how, perhaps because Figure 4C is so vague and incomprehensible.

DNA translocation during loop extrusion. We have modified the Discussion section to clarify this point.

6a) Is K363 on Scc1 important for the DNA binding ability of the complex? It would also be useful to highlight this residue on Figure 2. This experiment is required to confirm the in vivo relevance of the enhancement of DNA association by the Scc1 fragment in the in vitro experiments.6b) Scc1 is clearly required for the DNA-binding activity of Scc3-Scc1. The structure suggests that Scc1 K363 might contact DNA. Does Scc1 K363E reduce the binding of Scc3-Scc1? Related to this, even though the authors cannot see any density of the N-terminal region of Scc1K, this region might contribute to DNA binding. This should be experimentally tested. In the human SA2-Scc1K structure, the corresponding region in Scc1K forms a helix that is located at the base of the "nose" of SA2. Can the authors build a model of the SA2-Scc1-DNA complex and see if this N-terminal region of Scc1K might be close to DNA?NOTE: Concern on the function of K363 was raised by two reviewers and is reported here in its original wording as points 6a and 6b.

To address questions 6a and 6b, we have mutated Scc1 residue K363, as well as the neighboring R364, to arginine. We have also updated Figure 2 to highlight residue K363, as requested.

We found that Scc3-Scc1 complexes with the K363E/R364R double mutation display a DNA equilibrium dissociation constant that is essentially indistinguishable from that of the wild-type complex. We have added a description of these data in the Results section et seq. and updated Figure 3C accordingly. As we observed no phenotype of this mutation in vitro, we anticipate that yeast cells that express this mutant version of Scc1 would most likely show no advert phenotype, taking into account that even cells that express individual Scc3 patch mutants are perfectly viable (Figure 3E). We therefore did not further test this mutant Scc1 in vivo.

To further test the possibility that the disordered N-terminus might contribute to DNA binding, we removed the disordered region of Scc1K spanning amino acids 301–354. Unfortunately, the truncated Scc1K fragment (amino acids 355–400) failed to co-purify with Scc3. As shown in the SDS-PAGE gel (Author response image 1), we did not detect a band for the truncated Scc1K at the expected position, even when we highly overloaded the gel. Our interpretation is that the disordered region of Scc1K might contribute to the solubility of this fragment under overexpression conditions. This prevented us from testing DNA binding of a minimal Scc3T/Scc1 complex to investigate whether the disordered segment of Scc1K contributes to DNA binding.

In an alternative attempt to address whether the missing region of Scc1 might potentially interact with DNA, we built a composite model of the SA2-Scc1-DNA complex, as suggested by the reviewers. As shown in Author response image 2, the DNA is principally accommodated in the DNA binding ‘cradle’ of SA2 in this model. The N-terminal region of Scc1, including the mentioned a helix_334-342_, is clearly located too far away to engage in direct DNA contacts. Thus, while we cannot exclude the possibility that Scc1 topologically embraces DNA, similar to the situation in the condensin Ycg1-Brn1 subcomplex, there is no evidence that it directly contributes to the DNA-binding activity of Scc3-Scc1.

**Author response image 2. respfig2:** 

7) The ChIP-qPCR is essential for the conclusions of the paper but there are some issues with the presented experiment in Figure 3D. A minimum of 3 biological repeats are required to compute standard deviation, so the error bars here are not appropriate and should be removed. The authors could show the data for the two biological replicates side by side without error bars as an indication of reproducibility or, better, repeat the experiment a third time and calculate standard deviation. What do the percentages mean above the bars? How did the authors analyse the heptamutant given that it is not viable? Do the cells also carry endogenous Scc3? In this case, do all the strains in this experiment carry untagged Scc3 in addition to the tagged wild type or mutant protein? The fact that patch 2 mutants still bind DNA in vitro predicts that the patch 2 mutant protein should also associate with the chromosome in vivo, but this was not tested.

As requested, we are now showing the original data points in a revised version of the graph. The effects of the patch 3 mutation and heptamutant on condensin binding to chromosomes are very clear and consistent over all loci tested, and we are convinced that this wouldn’t change in a third biological repeat.

To be able to measure also nonviable mutants, we performed the ChIP experiments in diploid yeast strains that express an ectopic copy of Scc3-PK_6_ (including the wild-type control) and an untagged wild-type copy of Scc3 from one of its endogenous alleles (see detailed genotypes in Supplementary file 3).

To test the relevance of the DNA binding site revealed in our crystal structure, we decided to test one mutant that reduces (patch 3) and one mutant that largely abolishes (heptamutant) the positive charge in the DNA binding groove for ChIP experiments. Since the in vivo chromosome-binding data excellently correlates with the in vitro DNA-binding data (compare Figure 3C and 3D), we are convinced that the data presented in this figure are reasonably representative. The central conclusion of these data is that integrity of the DNA-binding surface of Scc3 is important for the association of cohesin with chromosomes. In our view, testing patch 1 or 2 mutant data in vivo would not provide any major additional insights.

8) The authors are proposing that the reported interactions are required for cohesin loading. However, an alternative possibility is that "core" cohesion can load but that Scc3 fails to associate with it. This should be tested by assessment of the association of other "core" cohesin subunits (Smc1/Smc3/Scc1) with chromosomes in the patch 3 mutant cells.

Previous work has shown beyond doubt that chromosome association of the other three core cohesin subunits strictly depends on the Scc3 subunit (Toth et al., 1999). Furthermore, we show that heptamutant and wild-type Scc3 bind Scc1 to a similar degree (Figure 3C). By extension, they must be incorporated into cohesin complexes in yeast cells with equal efficiency, indicating that mutagenesis of the DNA-binding interface does not perturb recruitment of Scc3 to cohesin.

9) What is the effect of the observed mutations on sister chromatid cohesion? The authors should test this using the TetR-GFP or LacI-GFP system.

The ability of cells to divide (Figure 3E) is an excellent read-out for the capacity of cohesin mutants to generate sister chromatid cohesion. As none of the three mutant patches alone has any obvious impact on cell division, these cells must be able to generate sister chromatid cohesion to a degree that enables efficient chromosome segregation. Hence, the combination of viability assays and ChIP-qPCR experiments provide adequate biological context to the structural biochemistry presented in this manuscript.

We would also like to draw the reviewers’ attention to another recent study (Elbatsh et al., 2016), which demonstrated that cohesin levels can be reduced quite significantly (to an extent mirrored by the patch 3 mutant) in human cells without any obvious impact on sister chromatid cohesion, consistent with previous work in budding yeast (Heidinger-Pauli et al., 2010).

Since complexes harbouring the heptamutant version of Scc3 fail to load onto chromosomes (Figure 3D), they by definition cannot generate sister chromatid cohesion and consequently cells fail to divide.

10) Hara et al. (2014) showed that mutating conserved basic residues in the N-terminal and middle regions of Scc1K did not affect Scc1K binding to SA2. These regions may transiently dissociate from SA2/Scc3 while the C-terminal region of Scc1K is still anchored to SA2/Scc3. It is thus possible that Scc3-Scc1K form a topological embrace of DNA, similar to the condensin sub-complex. This possibility needs to be discussed, especially if the N-terminal region of Scc1K is required for DNA binding (see point 6b).

Considering that Scc1 is clearly required for DNA binding, such a scenario is of course plausible. We now explain in reference to condensin (Discussion section et seq.): ‘The relevant section of Scc1 that would contribute to such topological DNA entrapment is disordered in our structure. As Scc1 is clearly required for DNA binding of the Scc3-Scc1 subcomplex but not through direct DNA interactions (Figure 3C), it is possible that cohesin uses a similar mode of chromatin engagement.’

[Editors' note: further revisions were requested prior to acceptance, as described below.]

I have now examined your resubmission and I am happy to inform you that I consider it essentially ready for acceptance. However, before formal acceptance, I would like to note the following three points:1) You seem to be using two somewhat different color schemes for Scc3 in the different figures, more bluish in Figure 2 and Figure 4, and more violet in Figure 1. May I suggest that you make the colours more uniform?

I have revised Figure 1 accordingly. Colors are now matching.

2) In Figure 2—figure supplement 1E, the right hand panel appears to be a composite of pasted lanes. If this is the case, could you please clearly mark this on the figure with a black vertical line and add a short reference to lane pasting in the legend?

Indeed, Figure 2—figure supplement 1E is a composite figure as non-relevant lanes have been cropped out. I have added a black vertical line and mentioned the following in the figure legend: “For the gel showing WaplFL, the black vertical line indicates the position where the gel has been cropped.”